# Fat and Carbohydrate Interact to Potentiate Food Reward in Healthy Weight but Not in Overweight or Obesity

**DOI:** 10.3390/nu13041203

**Published:** 2021-04-06

**Authors:** Emily E. Perszyk, Zach Hutelin, Jessica Trinh, Arsene Kanyamibwa, Sophie Fromm, Xue S. Davis, Kathryn M. Wall, Kyle D. Flack, Alexandra G. DiFeliceantonio, Dana M. Small

**Affiliations:** 1Modern Diet and Physiology Research Center, New Haven, CT 06510, USA; emily.perszyk@yale.edu (E.E.P.); zach.hutelin@yale.edu (Z.H.); jessica.trinh@yale.edu (J.T.); arsene.kanyamibwa@gmail.com (A.K.); sophie.fromm@charite.de (S.F.); xue.sun@yale.edu (X.S.D.); kathryn.wall@yale.edu (K.M.W.); 2Department of Psychiatry, Yale University School of Medicine, New Haven, CT 06511, USA; 3Department of Dietetics and Human Nutrition, College of Agriculture, Food, and Environment, University of Kentucky, Lexington, KY 40508, USA; kyle.flack@uky.edu; 4Department of Human Nutrition, Foods and Exercise, Virginia Tech, College of Agriculture and Life Sciences, Blacksburg, VA 24061, USA; dife@vtc.vt.edu; 5Center for Transformative Research on Health Behaviors, Fralin Biomedical Research Institute at Virginia Tech, Roanoke, VA 24016, USA; 6Department of Psychology, Yale University, New Haven, CT 06511, USA

**Keywords:** food reward/reinforcement, willingness to pay, macronutrient, carbohydrate, fat, BMI

## Abstract

Prior work suggests that actual, but not estimated, energy density drives the reinforcing value of food and that energy from fat and carbohydrate can interact to potentiate reward. Here we sought to replicate these findings in an American sample and to determine if the effects are influenced by body mass index (BMI). Thirty participants with healthy weight (HW; BMI 21.92 ± 1.77; *M ± SD*) and 30 participants with overweight/obesity (OW/OB; BMI 29.42 ± 4.44) rated pictures of common American snacks in 120-kcal portions for liking, familiarity, frequency of consumption, expected satiety, healthiness, energy content, energy density, and price. Participants then completed an auction task where they bid for the opportunity to consume each food. Snacks contained either primarily carbohydrate, primarily fat, or roughly equal portions of fat and carbohydrate (combo). Replicating prior work, we found that participants with HW bid the most for combo foods in linear mixed model analyses. This effect was not observed among individuals with OW/OB. Additionally, in contrast with previous reports, our linear regression analyses revealed a negative relationship between the actual energy density of the snacks and bid amount that was mediated by food price. Our findings support altered macronutrient reinforcement in obesity and highlight potential influences of the food environment on the regulation of food reward.

## 1. Introduction

Food choices depend on a sophisticated interaction of biology with the social, economic, perceptual, and nutritional characteristics of food. Biological signals conveying nutritional information may be conscious, such as the sweetness of watermelon, or they may be unconscious, as in the case of peripherally-derived signals generated during nutrient metabolism, such as glucose oxidation [1]. Understanding how these conscious and unconscious signals are integrated to regulate food choice may reveal new insights into the mechanisms by which the modern food environment promotes overeating. For example, processed foods are specifically associated with a number of deleterious health outcomes including not only obesity and diabetes [2,3], but also depression [4], cardiovascular disease [5], and all-cause mortality [6]. Processed foods are also characterized by unique nutritional characteristics not previously encountered in our evolutionary past [7]. Artificial sweeteners, for example, provide sweet taste unaccompanied by energy, a decoupling proposed to dysregulate metabolism [8,9,10,11]. Moreover, processed foods tend to be more energy-dense and this is associated with a worsened ability to estimate expected satiety [12]. Brain food cue reactivity is also potentiated for energy-dense foods compared to foods with low energy density, which are also perceived as healthier [13,14].

In the current study, we focus on the tendency of processed foods to contain large amounts of fat and carbohydrate in single products, a rare combination in the natural food environment [7]. Accumulating evidence suggests that energy signals for fat and carbohydrate engage separate gut-brain pathways [15,16,17,18,19,20,21,22] that can interact to potentiate reward (i.e., the “supra-additive effect”) [1,23,24]. Preclinical models of diet-induced obesity show that diets containing fat and carbohydrate are more effective at producing weight gain when both macronutrients are offered rather than when only one is available [25]. Likewise, DiFeliceantonio and colleagues quantified food reward in humans using an auction task where participants bid against a computer for the opportunity to eat a range of snack foods [26]. They demonstrated that people with healthy weight (HW) are willing to pay more for foods containing a combination of fat and carbohydrate (combo) than for equally liked and energetic foods containing predominantly fat or carbohydrate alone [23]. Responses in key brain reward circuits, including the striatum, were also shown to vary more closely with bid amount for combo foods compared to those with primarily fat or carbohydrate [23]. If distinct signals for fat and carbohydrate exert a supra-additive effect on reinforcement as prior findings suggest, then the frequency of this macronutrient combination in processed foods may be one mechanism that promotes their consumption.

Key to understanding the mechanism behind this supra-additive effect is the discovery that signals generated in the periphery during food consumption and metabolism play a critical role in determining reinforcement [1]. For example, rodents will self-stimulate for optogenetic activation of brainstem neurons that receive vagal afferent signals generated from the intestine [15]. Moreover, post-oral signaling appears to operate independently of conscious signals such as beliefs about energy content and perceived liking in humans [1,27]. Conditioned striatal and hypothalamic responses to energy-predictive cues are strongly correlated with elevations in blood glucose during energy consumption, but not with the rated liking of the beverages providing the energy [27]. Likewise, the amount of money participants with HW are willing to pay for food items during an auction task correlates with actual, and not estimated, energy density [23,28]. This evidence highlights that implicit signals about the energetic properties of foods regulate central reward circuits independently of explicit signals, paving the way for these implicit signals to combine across macronutrients and potentiate reward. In doing so, organismal behaviors would be biased towards the consumption of high energy-dense foods offering multiple energy sources compared to a single one.

Nonetheless, we suspect that the nature of the supra-additive effect may differ as a function of Body Mass Index (BMI). A greater motivation to consume such palatable, energy-dense foods that does not habituate over time is often exhibited by individuals with obesity compared to HW [29,30,31]. It is therefore possible that the reinforcing value of foods containing both fat and carbohydrate is further elevated in obesity. However, obesity is also frequently accompanied by an insensitivity to post-oral signaling [32,33], likely consequent to poor diet [24]. The combination of fat and carbohydrate may promote overeating in HW, but as dietary habits change and reward systems adapt, specific macronutrient effects could become blunted or difficult to detect. Since the consumption of processed foods is positively associated with BMI [3,34], and global food reinforcement is typically greater [29,30], macronutrient effects on food reinforcement could instead be absent in people with excess weight.

The present study is the first to examine whether the potentiation of fat and carbohydrate on willingness to pay (WTP) for snack foods is amplified or attenuated in obesity. While testing these competing hypotheses about the role of BMI, we also aimed to determine if we could replicate prior reports (1) of the supra-additive effect in HW [23], and (2) that actual, but not estimated, energy density drives food reinforcement [23,28]. Whereas the previous studies were conducted in Canada [28] and Germany [23], our study was performed in the United States (US). Processed foods high in both fats and carbohydrates make up a greater proportion of total energy intake in the US compared with Germany or Canada [3,35,36]. This raises the possibility that Americans may not exhibit the supra-additive effect and/or that the association between actual versus estimated energy density and WTP, a measure of food reinforcement, might differ.

## 2. Materials and Methods

### 2.1. Participants

Participants were recruited for the study from the greater New Haven, Connecticut area by flyer or social media advertisements. Participants were required to be fluent in English and have lived in the United States for the majority of the last five years without any interruption longer than nine months. This restriction was set to maximize prior exposure to American snack foods since the supra-additive effect is expected to be based on prior associations with implicit post-oral nutritional signals. Participants could not be currently dieting or have any dietary restrictions or food allergies to the items in our study. Additional exclusion criteria included: (a) current or past psychiatric illness, (b) medications or psychoactive drugs that could affect alertness during testing, (c) known taste or smell dysfunction, (d) a diagnosis of diabetes or hypertension, (e) current pregnancy/nursing, and (f) other serious medical conditions such as cancer, and (g) advanced degree training in nutrition. Eligibility was determined using an online screening form and follow-up email correspondence. Additionally, participants completed the Beck Anxiety Inventory (BAI) [37] and the Beck Depression Inventory II (BDI) [38] in order to confirm that they did not meet criteria for clinically significant depression or anxiety [39,40]. All participants provided written informed consent and study procedures were approved by the Yale Human Investigations Committee.

To determine sample size for our primary goal of replicating macronutrient effects on food reinforcement, a power analysis was conducted with G*Power 3 [41] using previously collected data from the auction task [23]. We first calculated the effect sizes from the means and standard deviations of the differences in WTP as d = 1.103 for combo versus carbohydrate foods and as d = 0.536 for combo versus fat foods from the German study. For a conservative estimate using the smaller effect size, we determined that a total sample of 30 participants would be needed for a two-tailed t-test of two dependent means at an alpha of 0.05 to achieve a power of 0.80. We therefore aimed to include *n* = 30 participants with HW (defined as BMI < 25 kg/m^2^) as well as *n* = 30 with overweight/obesity (OW/OB; BMI ≥ 25 kg/m^2^) in order to test the role of BMI. Recruitment into the low and high BMI groups was counterbalanced to minimize demographic differences between them. More specifically, age and household income were stratified such that randomization was broken in order to include participants of older age and with lower household income into the HW group. A total of five participants were excluded from final analyses (see Section 2.5). We therefore recruited an additional five participants to achieve *n* = 30 in each BMI group.

### 2.2. Food Pictures 

MacroPics is a 36-item picture set of American snack foods [42]. Each food image portrays a 120-kcal portion that is classified into one of three categories by macronutrient content: (1) predominantly carbohydrate, (2) predominantly fat, (3) or a combination of carbohydrate and fat (combo) [42]. Food images in these carbohydrate, fat, and combo categories differ minimally in a number of visual properties such as color and intensity, as well as in objective qualities (e.g., food energy density, price, and sodium content) and subjective (e.g., perceived liking, familiarity, and estimated energy content) characteristics using ratings provided by 128 participants with a range of BMIs in a prior study [42]. Examples of the MacroPics stimuli are presented in Figure 1a. 

### 2.3. Measures

#### 2.3.1. Internal State

Current hunger, fullness, thirst, potential to eat, and desire to eat (Table 1) were rated on continuous horizontal 260-mm visual analog scales (VAS) fit to the size of the computer window. Each internal state rating was scored from 0–100 as the percentage of length along the line.

#### 2.3.2. Subjective Ratings of the Food Pictures

Participants were asked to use rating scales to indicate liking, familiarity, frequency of consumption, expected satiety, healthiness, estimated energy content, estimated energy density, and estimated price. The majority of these variables were selected for consistency with prior work [23]. We included the additional variables of frequency of consumption, expected satiety, and estimated price as they may influence the reinforcing value of food [43,44,45,46]. All ratings were acquired using PsychoPy version 3.0 [47] and details of the rating scales used are provided in Table 2. Liking was assessed with the category-ratio Labeled Hedonic Scale depicted on a 150-mm vertical line and scored from −100 (maximal disliking) to +100 (maximal liking) based on percentage of deviation from zero along the VAS [48]. All other variables were measured using a continuous horizontal 260-mm VAS. Frequency was scored in days per month, estimated energy content in kcal, and estimated price in USD. All other ratings were scored from 0–100 as the percentage of length along the VAS line.

#### 2.3.3. Eating Behavior

Self-reported restrained, emotional, and external eating behaviors were measured with the Dutch Eating Behavior Questionnaire (DEBQ) [49]. The DEBQ has been validated in English-speaking populations [50] and shows internal consistency at ICC ≥ 0.79 [49].

#### 2.3.4. Dietary Consumption of Fat and Sugar

Participants completed a modified version of the Dietary Fat and Free Sugar Short Questionnaire (DFS) [51]. The original DFS was created for use in Australia; here we tailored the questions for Americans. Food names were modified from common Australian foods/terminology to the corresponding American foods/words. For example, the item “fried chicken or chicken burgers” was changed to “fried chicken or chicken wings/tenders/nuggets.” In the DFS, participants report the frequency of monthly intake of 26 selected fatty and sugary foods from 1–5, with larger scores representing greater consumption. Fat, sugar, and fat plus sugar (fat-sugar) subscores are obtained from summing responses to the 11, 9, and 6 items in each respective group. The total sum is also computed for a composite DFS score. We chose this metric to assess whether the frequency of consumption of fat, carbohydrate, and combo foods differed as a function of BMI.

#### 2.3.5. Anthropometric Measures

Participant waist and hip circumferences were measured to the nearest centimeter (cm) using a tape measure in order to compute waist-hip ratio (WHR). Height was measured to the nearest cm with a digital stadiometer and weight to the nearest hundredth of a kilogram (kg) with an electronic scale. BMI was calculated as weight divided by the square of height (in kg/m^2^) to the nearest hundredth. Bioelectric impedance analysis (Seca Medical Body Composition Analyzer mBCA 515, Hamburg, Germany) was used to obtain body fat percentage to the nearest tenth of a percent.

#### 2.3.6. Auction Task

Participants completed four blocks of the Becker–DeGroot–Marshak auction task [26]. In this task, participants bid up to 5 USD each for snack foods on a 260-mm VAS with labels for 0, 2.50, and 5 USD. They were told that they were competing against a computer to win snack foods, and that a random trial would be selected at the end of the task. If, on that trial, their bid was greater than the computer’s bid, then they would receive the snack item in exchange for their bid amount. If, however, their bid was lower than the computer’s bid, they would receive the 5 USD and no snack. Unknown to the participants, the selected trial was restricted to one of four shelf-stable foods kept in the lab: fruit snacks, cheese and crackers, peanut butter and crackers, and Doritos chips. Each block consisted of 36 trials presented randomly and separated by a jittered ~4 s inter-trial interval where a fixation cross was shown (Figure 1b). In each trial, one of the 36 MacroPics food items was displayed for 4 s, and participants then had 5 s to bid for that food.

### 2.4. Procedure

The study consisted of two session days. The majority of sessions took place between 12pm and 6pm and lasted ~1 h. On day 1, participants were instructed to arrive at least 1-h fasted. To start, participants were trained to use the rating scales (Table 2). They were given definitions of a “calorie” as “a unit of energy,” of “energy content” as “the amount of energy people get from the foods or drinks they consume,” and of “energy density” as “the number of calories stored in the food per unit volume.” They were also given an example of the difference between energy content and energy density and were allowed to ask questions for clarification. Participants then made subjective ratings (Table 2) of the 36 MacroPics food images in randomized order. Next, participants provided general demographic information and completed computerized versions of the BAI, BDI-II, DEBQ, and DFS. Finally, anthropometric measurements were obtained. Participants were compensated 30 USD in cash upon completion of the first session.

On day 2, participants were instructed to arrive in a hungry state and at least 4-h fasted. After providing internal state ratings (Table 1), participants completed the auction task to bid up to 5 USD for each food item. Afterward, a computer bid from 0–5 USD was randomly generated for the selected trial. Participants were given time to consume their snack if won; 15% of participants obtained a snack by outbidding the computer on the selected trial. Lastly, participants rated what they believed the grocery store price of each item would be (“estimated price” in Table 2). All participants were compensated with 20 USD in addition to their earnings from the auction task (up to 5 USD) upon completion of the second session.

### 2.5. Statistical Analyses and Data Visualization

Data processing, general linear models (GLMs), linear mixed models (LMMs), ANOVAs, Student’s t-tests, linear regressions, and Pearson correlations were performed in Matlab R2018b (The MathWorks Inc., Sherborn, MA, USA). The code used to generate the results is available upon request. Data were plotted in GraphPad Prism version 8 (GraphPad Software, La Jolla, CA, USA). Because participants completed four blocks of the auction task, their WTP for each food item was calculated as the average of their individual bids across those four blocks. Prior to formal analysis, data from participants who bid an average of 0 USD on >20 food items were removed. Data from 5 of the initial 65 participants were excluded on this basis. Alpha was set to 0.05 for all analyses. Corrections for multiple comparisons were made by adjusting the alpha threshold for the number of tests at each step using the Bonferroni method. Pre-planned analyses included: (1) assessing the main effects and interactions of macronutrient category and BMI group on WTP using LMMs along with pairwise comparisons between fat, carbohydrate, and combo categories; (2) performing follow-up t-tests to examine differences between BMI groups in WTP, food liking, DFS score, and internal state; and (3) testing for associations among actual energy density, estimated energy density, and WTP for all stimuli, as well as within each macronutrient category using averages per food item across participants with HW and OW/OB. Follow-up regressions and mediation analyses to explore the role of food price in bidding behavior [52] were unplanned.

## 3. Results

### 3.1. Participant Characteristics

Demographic and anthropometric characteristics of the final sample are described in Table 3. No significant differences were observed between the HW and OW/OB groups in age (t_(1, 58)_ = 0.981, *p* = 0.331), education (t_(1, 58)_ = 0.662, *p* = 0.511), or household income (t_(1, 58)_ = 1.861, *p* = 0.068) using unpaired, two-sample *t*-tests. In the HW group, 19 participants identified as White, 8 as Asian, and 3 reported having more than one race. In the OW/OB group, 20 identified as White, 1 as Black or African American, 2 as Asian, 3 as more than one race, and 4 preferred not to report. There were 5 participants who reported being Hispanic or Latinx in each BMI group (and 25 non-Hispanic or Latinx).

### 3.2. Macronutrient Content Impacts WTP for Food in Participants with HW, but Not with OW/OB

We first tested whether macronutrient category and BMI group (<25 or ≥25) influence WTP for foods in our American sample. Guided by the analyses used in our earlier study of the effect of macronutrient on bidding behavior (i.e., WTP) in German participants with HW [23], we ran a LMM across all participants with bid/WTP as the outcome variable; BMI group, macronutrient category, the interaction of BMI group × macronutrient category, actual energy density, estimated energy density, estimated energy content, portion size, and liking as fixed effects; and participant as a random effect. We found a significant main effect of macronutrient category (F_(2, 2149)_ = 5.733, *p* = 0.003) and a significant interaction of BMI group × macronutrient category (F_(2, 2149)_ = 3.142, *p* = 0.043) on WTP, but no main effect of BMI group (F_(2, 2149)_ = 0.524, *p* = 0.469; Figure 2a). Follow-up pairwise comparisons revealed no significant differences in WTP as a function of BMI group for fat (t_(1, 713)_ = 0.860, *p* = 0.390), carbohydrate (t_(1, 713)_ = 0.726, *p* = 0.468), or combo foods (t_(1, 713)_ = 0.130, *p* = 0.897). Rather, the BMI × macronutrient category interaction emerged because there was a main effect of macronutrient category on WTP among participants with HW (F_(2, 1072)_ = 6.661, *p* = 0.001), but not among individuals with OW/OB (F_(2, 1072)_ = 1.905, *p* = 0.149; Figure 2a). This result among participants with HW was driven by greater bids for combo versus carbohydrate foods (t_(1, 713)_ = 3.595, *p* < 0.001), but not for combo versus fat (t_(1, 713)_ = 1.055, *p* = 0.292) or carbohydrate versus fat (t_(1, 713)_ = 2.068, *p* = 0.039) items after correction for multiple comparisons.

Importantly, these differences in the impact of macronutrient content on WTP across BMI groups were independent of differences in mean liking (t_(1, 58)_ = 0.002, *p* = 0.998) or overall average bid for all foods (t_(1, 58)_ = 0.703, *p* = 0.478) between participants with HW (liking 0.56 ± 0.14; bid 0.99 ± 0.59; *M*
±
*SD*) and participants with OW/OB (liking 0.57 ± 0.10; bid 1.10 ± 0.68; *M*
±
*SD*). Furthermore, we found no interaction (F_(2, 2154)_ = 0.361, *p* = 0.697) or main effects of BMI group (F_(1, 2154)_ = 0.210, *p* = 0.647) or macronutrient category (F_(2, 2154)_ = 0.863, *p* = 0.422) on liking ratings (Figure 2b) using a LMM with liking as the outcome variable; BMI group, macronutrient category, and the interaction of BMI group × macronutrient category as fixed effects; and participant as a random effect. Descriptive statistics for subjective ratings and WTP for foods across all macronutrient categories are provided in Appendix A for the HW group and Appendix A for the OW/OB group. To further rule out a role for hunger or dietary habits in our BMI group × macronutrient category interaction on bidding behavior, we compared internal state ratings and DFS scores of HW and OW/OB groups. No significant differences between BMI groups were found for DFS composite score (t_(1, 58)_ = 0.255, *p* = 0.800) or any DFS subscore or internal state rating (Appendix A).

### 3.3. Fat and Carbohydrate Potentiate WTP for Food in Participants with HW

Our next step was to assess whether we were able to replicate the supra-additive effect found previously in the German sample with HW [23]. We created an additional LMM with the same covariates, but in which foods were categorically coded as containing fat or carbohydrate, to test for an interaction of fat × carbohydrate on WTP in HW. As previously observed [23], the effect of combining fat plus carbohydrate on WTP was supra-additive (F_(1, 1072)_ = 9.187, *p* = 0.002) in the HW group. Additionally, to improve upon this test of supra-additivity, we also recoded each food by the actual quantity of fat or carbohydrate it contained in g/120 kcal and tested for the interaction of fat × carbohydrate on bids. Once again, we observed that fat and carbohydrate potentiate food reinforcement in our American participants with HW (F_(1, 1071)_ = 4.746, *p* = 0.030).

### 3.4. Associations among WTP and Actual and Estimated Energy Density

Our second aim was to investigate whether WTP is positively associated with implicit energy signals rather than explicit judgments of energy density as in prior work [23,28]. To this end, we performed linear regressions between WTP and actual and estimated energy density. We first computed separate averages per food item for participants with HW and with OW/OB, allowing us to include an interaction term for BMI group in our GLMs. Across all food stimuli and within each individual macronutrient category, there was no significant interaction of BMI group in the relationships between actual energy density and WTP, estimated energy density and WTP, or actual and estimated energy density (Appendix A). We therefore collapsed across BMI group to compute a single average for each food item from all participants for the remaining regressions.

Consistent with prior reports [23,28], estimated energy density was not associated with WTP (Figure 3a). However, we unexpectedly found that WTP was negatively associated with actual energy density across all food items (r^2^ = 0.209, *p* = 0.005; Figure 3b). This result was driven by a strong effect in combo foods (r^2^ = 0.548, *p* = 0.006) that was absent in the carbohydrate and fat categories (Figure 3b). Though not significant, we also observed a weak negative relationship between estimated and actual energy density across all food items (r^2^ = 0.099, *p* = 0.062). When tested separately in each macronutrient category, this negative association was significant for combo (r^2^ = 0.587, *p* = 0.004), but not carbohydrate or fat, foods. Therefore, participants tended to underestimate the energy density of snacks in the combo category.

### 3.5. Food Price Mediates the Negative Relationship between Actual Energy Density and WTP

We aimed to better understand the unexpected negative relationship between actual energy density and WTP. As this effect was driven by a strong association among combo foods, we wanted to determine if there was a third variable related to energy density in this macronutrient category (but not in the fat or carbohydrate categories) that helped to account for our results. Using only combo food items, we first correlated actual energy density with all remaining food characteristics and subjective ratings (Appendix A). Actual energy density was significantly related to volume, actual price, and participant ratings of estimated price, expected satiety, and healthiness after correction for multiple comparisons at this step (Appendix A). Using all food items, we then performed an ANOVA to assess the interaction of each of these variables with macronutrient category on actual energy density. We observed strong macronutrient interactions with actual volume (F_(2, 25)_ = 5.098, *p* = 0.014) and actual price (F_(2, 25)_ = 8.869, *p* = 0.001), and a weak interaction with expected satiety (F_(2, 25)_ = 3.379, *p* = 0.050). In follow-up comparisons, we found that the impact of actual price on actual energy density was significantly different for carbohydrate versus combo (t_(1, 20)_ = 3.301, *p* = 0.004) and fat versus combo (t_(1, 20)_ = 3.662, *p* = 0.002), but not carbohydrate versus fat (t_(1, 20)_ = 0.376, *p* = 0.711), items. Indeed, actual energy density was related to actual price across all food items (r^2^ = 0.417, *p* < 0.001) and in the combo category (r^2^ = 0.859, *p* < 0.001), but not in the carbohydrate or fat categories (Figure 4a). The associations between energy density and volume or expected satiety did not follow this pattern (Appendix A), indicating that they were unlikely to account for the strong negative association between energy density and WTP specific to the combo category.

We therefore reasoned that price could be a likely candidate to explain the strong negative association observed between WTP and actual energy density among combo, but not carbohydrate or fat, snacks. To test this, we first confirmed that actual price was positively associated with bidding behavior across all food stimuli (r^2^ = 0.352, *p* < 0.001) and in the combo category (r^2^ = 0.748, *p* < 0.001; Figure 4b). We then performed a formal mediation analysis [52] and found that food price fully mediates the relationship between WTP and actual energy density across all food items (Figure 5a) and within the combo category (Figure 5b). After identifying the role of food price in bidding behavior, we sought to ensure that the supra-additive effect of fat and carbohydrate on WTP that we observed in participants with HW would remain significant after including actual price as a covariate. We tested the same LMMs as before (see Section 3.2), but with food price added as a fixed effect. The interaction of fat × carbohydrate on participant bids remained significant regardless of whether we coded food items categorically (F_(1, 1071)_ = 6.555, *p* = 0.011) or by the actual grams of fat or carbohydrate they contained (F_(1, 1070)_ = 5.825, *p* = 0.016).

Each food item in our American picture set (MacroPics) contained precisely 120 kcal [42]. We therefore wanted to verify that our inability to replicate the positive association between WTP and energy density from prior reports [23,28] was not due to a lack of variance in energy content. To this end, we employed a modified version of the MacroPics stimuli [42]. Images now depicted 40, 120, or 200-kcal portions with the same variation across macronutrient categories (i.e., four images each of 40, 120 or 200 kcal portions; Figure 6). We recruited an additional 22 naïve participants with HW to serve as an independent sample to rate and bid for the foods shown in this modified picture set. We kept all other procedures the same. Participant characteristics of the independent cohort are provided in Appendix A.

We again observed a negative association between WTP (for the portion shown) and actual energy density (r^2^ = 0.143, *p* = 0.023; Figure 7a). We also found that food price was negatively related to actual energy density (r^2^ = 0.109, *p* = 0.049; Figure 7b) and positively correlated with WTP (r^2^ = 0.389, *p* < 0.001; Figure 7c). Ultimately, our formal mediation analysis revealed that food price fully mediates the relationship between WTP and actual energy density across these food items in varying portions (Figure 7d). For full data visibility, we report fitted scatter plots comparing each food characteristic and subjective rating with actual energy density (Appendix A) and WTP (Appendix A) for foods pictured at 120 kcal. We provide the same for foods in varying portions rated by our independent sample (Appendix A).

## 4. Discussion

Our study had three main objectives. The first was to identify the influence of BMI on the supra-additive effect in which participants will pay more for equally liked, familiar, and energetic snacks containing both fat and carbohydrate compared to those containing primarily fat or carbohydrate alone [23]. The second and third were to determine if we could replicate prior work showing (1) the existence of the supra-additive effect in HW [23] and (2) that bid amounts increase for foods with greater energy density [23,28]. In line with previous research [23], we found that WTP was greatest for the combo foods and that calorie-for-calorie, combining fat and carbohydrate has a supra-additive effect on WTP in participants with HW. A key feature of MacroPics is that snack items in the three macronutrient categories do not significantly differ in cost [42], suggesting that food price did not impact these results. We also replicated the supra-additive effect after directly accounting for price. These findings build upon existing evidence that fat and carbohydrate interact to potentiate food reward [23]. In addition, we were the first to uncover that these effects were specific to the HW group, with macronutrient content exerting no significant influence on WTP in individuals with OW/OB. Finally, unlike in the prior Canadian [28] and German [23] samples, our American participants bid more for foods with lower energy density, and this negative relationship was mediated by food price.

### 4.1. Interaction of Carbohydrate and Fat on Food Reinforcement

Consistent with the German study [23], we identified a supra-additive interaction between fat and carbohydrate in determining WTP in participants with HW. Since other factors (e.g., liking and portion size) were accounted for in the statistical model, our findings are in agreement with prior work that macronutrient content contributes to the rewarding value of food in the context of the auction task [23]. However, we note that while the supra-additive interaction is reproducible and provides important insights into understanding the mechanisms underlying food reinforcement, its application to real world food choice may be limited when the effects of other factors are not controlled. Future studies designed to identify which of many variables (e.g., price, liking, energy density, macronutrient content) best predicts food choice are therefore needed.

Nevertheless, the current study provides additional evidence that the reinforcing signals generated in the periphery conveying nutritive information to the brain are distinct for energy from fat versus carbohydrate [1,15,16,17,18,19,20,21,22,23,24]. Our behavioral result is in agreement with classical work from the animal literature indicating that post-ingestive signals regulate flavor preference formation [53,54,55]. It further coincides with emerging findings that hepato-portal sensing of glucose oxidation is critical in driving dopamine release and reinforcement for glucose [16,21], whereas a peroxisome proliferator-activated receptor alpha (PPARα) dependent mechanism in upper intestine endocrine cells drives dopamine release and reinforcement for lipids [17]. Fat, but not sugar, also requires intact vagal signaling to inhibit the activity of hypothalamic agouti-related protein (AgRP)-expressing neurons [20] and promote feeding to satiation [19,22]. In contrast, responses to sugar are relayed to AgRP neurons via spinal afferents [20]. Accordingly, diets high in both fat and carbohydrate are more effective in stimulating overeating in rodents than those with only carbohydrate or fat [25,56,57]. Finally, our observation is in line with the report of a stronger association between thalamostriatal activity and bid amount for combo foods compared to those with primarily fat or carbohydrate [23].

### 4.2. Influence of BMI

In the current context, the success of the auction task is predicated on the assumption that the familiar food images represent conditioned stimuli that have well-learned associations with the post-oral reinforcing effects of the food items. Since obesity is often associated with greater reinforcement of energy-dense foods [29,30] but also perturbed central nervous system sensitivity to gut-derived signals [32,33], we reasoned that bidding behavior may differ in individuals with obesity. Supporting this hypothesis, macronutrient content did not influence WTP in participants with OW/OB. This result is consistent with evidence for impaired reinforcement learning and habituation to food in obesity across humans [58,59,60,61] and animals [56,62,63]. Likewise, many hormones—such as insulin—influence reinforcement [64], raising the possibility that metabolic dysfunction in obesity influences the effect of macronutrients on reward coding. We are also unable to rule out diet as a contributing factor. For example, it has been established that excess dietary fat may decrease taste sensitivity to fatty acids [65,66] and blunt dopamine signaling, potentially promoting further overconsumption of fat to restore its reinforcing value [17,67]. Evidence for similar modulatory effects of sweet intake on sweet taste intensity perception [68,69] and reinforcement [70] exists across organisms. In contrast, maintaining a low-sugar diet for one week dramatically increases the reinforcing value of sugary foods in people independent of BMI [71]. These effects are even present across nutrient types. Animals that spend more time on a high-fat diet lose motivation to obtain a sucrose reinforcer [72], and people who restrict intake of a target palatable food show reduced reinforcing value specific to that food [73]. Though typical consumption of fatty and sugary foods in the DFS did not significantly differ between our participants with HW and OW/OB, it remains unclear whether other measures would reveal discrepancies in diet.

### 4.3. Impact of the Food Environment

In both prior studies using the auction task to study food reinforcement, a positive correlation was found between actual, but not estimated, energy density and WTP [23,28]. This was interpreted as reflecting the importance of unconscious nutritional signals compared to conscious beliefs about nutrition in guiding food choice. These findings support the supra-additive macronutrient effect—which is independent of the perception of liking and estimated energy density and content—and are overall consistent with metabolic signals being the primary determinants of food reinforcement [1]. We were therefore surprised when the opposite relationship was observed; that is, in our US-based sample, we identified a negative association between actual energy density and WTP. There are several potential explanations for this unexpected result. First, all MacroPics food items contained precisely 120 kcal [42]. In the German study, the average energy content was ~128 kcal, but individual snack foods varied in energy within each macronutrient category [23]. The Canadian study also employed variable portions of both healthy, low energy-dense (i.e., fruits and vegetables) and unhealthy, high energy-dense (e.g., chocolate or chips) food items [28]. It is plausible that the lack of variance in energy content resulted in reduced sensitivity to detect an association between energy density and WTP. However, we ruled out this confounding variable by replicating the negative relationship between WTP and actual energy density when a separate cohort rated foods with varying portion sizes (40, 120, 200 kcal) in a modified version of the MacroPics picture set [42].

An alternative explanation could be that differences in diet across the German, Canadian, and American samples contribute to the discrepant results. Highly-processed foods make up a greater proportion of the typical diet in the US compared to Germany or Canada [3,35,36]. This is important because there is evidence that the consumption of processed foods impairs gut-brain signaling [24]. For example, mice fed a high-fat diet show blunted dopamine responses to intragastric lipid emulsion infusion [17]. Rodents with extended access to a Western diet rich in fat and carbohydrate also display reduced striatal D_2_ dopamine receptor expression and increased weight gain relative to those with restricted access [56]. Similarly, healthy humans fed ultra-processed compared to minimally-processed foods eat ~500 more calories per day over two weeks [74]. These findings indicate that the nutritional signals generated from eating processed foods under-represent the amount of energy intake. Thus, if habitual consumption of processed foods degrades the fidelity of post-ingestive energetic signals, then energy density would have a weaker association with reinforcement. If this were the case, then we should have observed no relationship rather than a strong negative association between WTP and energy density across all food items and particularly in combo foods.

Instead, we suggest that this negative association is most likely accounted for by the impact of food cost. We support this hypothesis by showing that food price mediated the negative relationship between WTP and actual energy density of all food items at 120 kcal, within the combo category itself, and even when food portion size was varied. There is a strong correspondence between food insecurity and obesity in the US [75], and the impact of socioeconomic status on food selection is a well-described phenomenon [46,76]. Thus, it is possible that the influence of price outweighed the ability for implicit signals of energy density to drive bidding behavior in our American sample. The previous German study did not control for food price in testing the relationship between energy density and WTP [23]. However, the Canadian study did, and found that the positive association between actual energy density and bidding behavior was still significant after including a covariate for price [28]. We believe that this is evidence for population-level differences between Germany, Canada, and the US that may encompass factors such as cost of living or disparities in the typical price of unprocessed versus processed foods. Future work directly comparing WTP or another metric of food reward with price across different cultures will be required to formally test these hypotheses.

### 4.4. Study Strengths, Limitations, and Future Directions

We improve upon previous investigations of nutrient composition and food reinforcement by adding measures for frequency of food consumption and expected satiety. To our knowledge, we are also the first to measure estimated price in conjunction with WTP, which was useful in testing whether actual or estimated price (in addition to volume, expected satiety, and healthiness) mediated the negative relationship between WTP and energy density. We further tested two variations of the MacroPics picture set [42] in participants over a range of BMIs from HW to OW/OB. We selected to use MacroPics because food items in the three macronutrient categories do not significantly differ over a large number of attributes (e.g., visual properties, energy density, price, and sodium content) [42]. However, the fat items do contain more protein (in g/120 kcal) than the carbohydrate and combo foods [42]. Therefore, we are unable to dissociate the effects of fat and protein within our current study. Future work would benefit from identifying the role of protein and using real food stimuli instead of images that rely on experience and learned responses. Enhanced methods of measuring the intake of fat and carbohydrate nutrients (e.g., 72-h dietary recalls) and of processed food products that improve upon our use of the DFS will also be important in disentangling the interactions of diet and food cost on the effects observed here. Likewise, the incorporation of neuroimaging techniques could confirm whether neural response when bidding for foods differs between participants with HW and OW/OB.

## 5. Conclusions

In summary, we demonstrate that the combination of fat and carbohydrate has a supra-additive effect on WTP among participants with HW, but not with OW/OB, as defined by BMI. We also highlight the importance of considering the role of the food environment in food choice. Finally, we speculate that common US diets made up of cheap, processed foods may disrupt the capacity of unconscious energy signals to drive food reinforcement.

## Figures and Tables

**Figure 1 nutrients-13-01203-f001:**
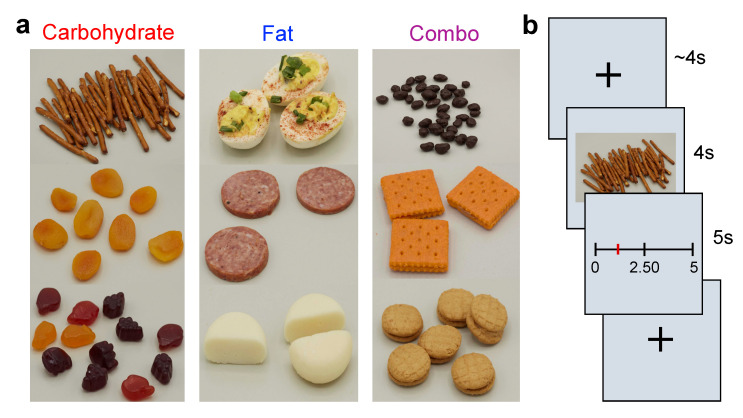
Study design. (**a**) Example MacroPics food pictures in the three macronutrient categories: carbohydrate, fat, and combo. (**b**) Sample trial from the auction task in which participants viewed a fixation cross, observed a food image, and bid for the snack from 0–5 USD.

**Figure 2 nutrients-13-01203-f002:**
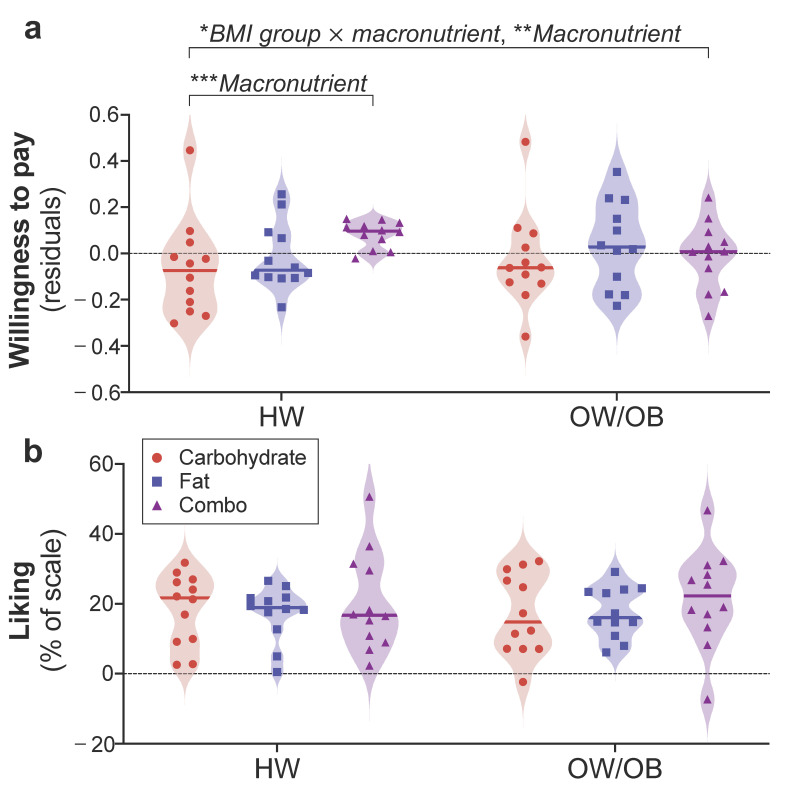
Effects of BMI group (HW, OW/OB) and macronutrient category (carbohydrate, fat, combo) on WTP and liking. (**a**) The interaction of BMI group × macronutrient category on WTP was significant (* *p* = 0.043), and there was an overall main effect of macronutrient (** *p* = 0.003) but no main effect of BMI group. This result was driven by a significant main effect of macronutrient category on WTP in the HW group (*** *p* = 0.001), but not in the OW/OB group. BMI groups did not differ in WTP for fat, carbohydrate, or combo foods. Residuals depict bids after accounting for covariates in the LMM: actual energy density, estimated energy density, estimated energy content, portion size, and liking. (**b**) No differences in the rated liking of the foods were observed across BMI group or macronutrient category, and there was no interaction of BMI group × macronutrient category on liking. Each data point depicts a single food item and shading represents the density of points around the median.

**Figure 3 nutrients-13-01203-f003:**
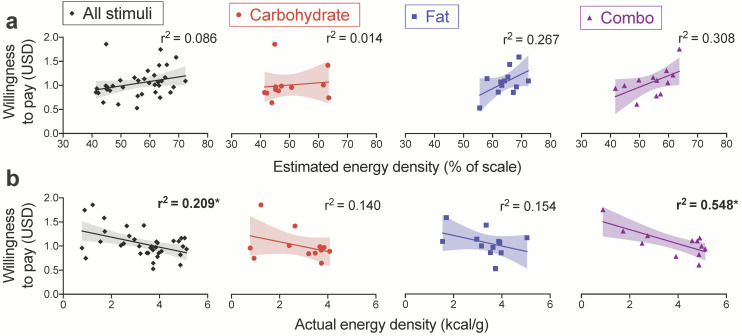
WTP is negatively associated with actual, but not estimated, energy density across all stimuli and in combo foods. Fitted scatter plots depict the relationships of WTP with (**a**) estimated energy density and (**b**) actual energy density. Resulting r^2^ values are shown for each regression performed on all food items (averaged across all participants) or broken down by the carbohydrate, fat, and combo categories. Each data point depicts a single item (36 total stimuli or 12 foods per macronutrient category) and shading indicates 95% CI. * *p* < 0.0063 after Bonferroni correction for the eight tests.

**Figure 4 nutrients-13-01203-f004:**
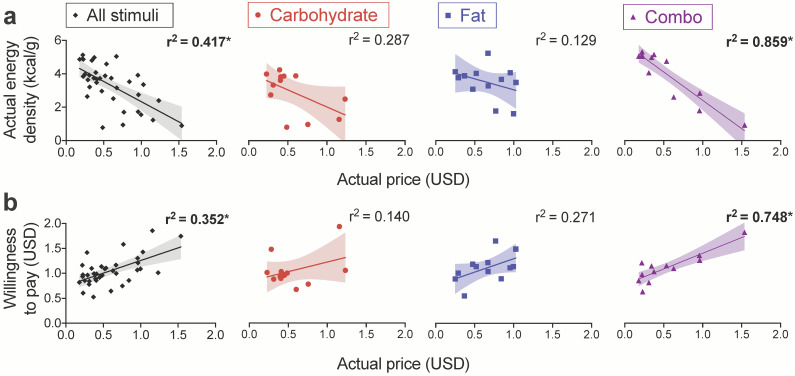
Actual price is associated with energy density and WTP across all stimuli and in combo foods. Fitted scatter plots depict the relationships of actual food price with (**a**) actual energy density and (**b**) WTP. Resulting r^2^ values are shown for each regression performed on all food items (averaged across all participants) or broken down by the carbohydrate, fat, and combo categories. Each data point depicts a single item (36 total stimuli or 12 foods per macronutrient category) and shading indicates 95% CI. * *p* < 0.0063 after Bonferroni correction for the eight tests.

**Figure 5 nutrients-13-01203-f005:**
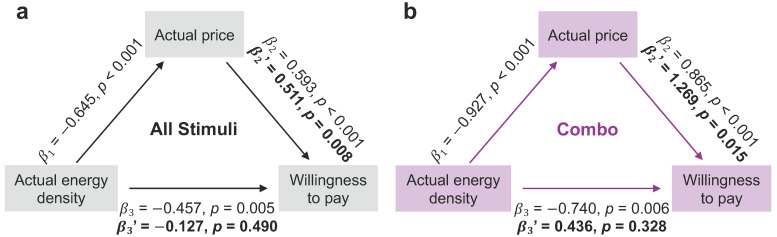
Food price mediates the negative association between energy density and WTP across all stimuli and in combo foods. (**a**) As energy density was separately associated with food price (β_1_) and WTP (β_3_), and price was also related to WTP (β_2_) across all stimuli, we tested whether price mediates the negative relationship between energy density and WTP. Indeed, the relationship between energy density and WTP *(*β_3′_*)* was no longer significant when the indirect effect of price was accounted for in the regression model, which itself remained significant *(*β_2′_*)*. (**b**) Actual food price also mediates the association between energy density and WTP when restricted to combo foods. Each β refers to a standardized beta coefficient.

**Figure 6 nutrients-13-01203-f006:**
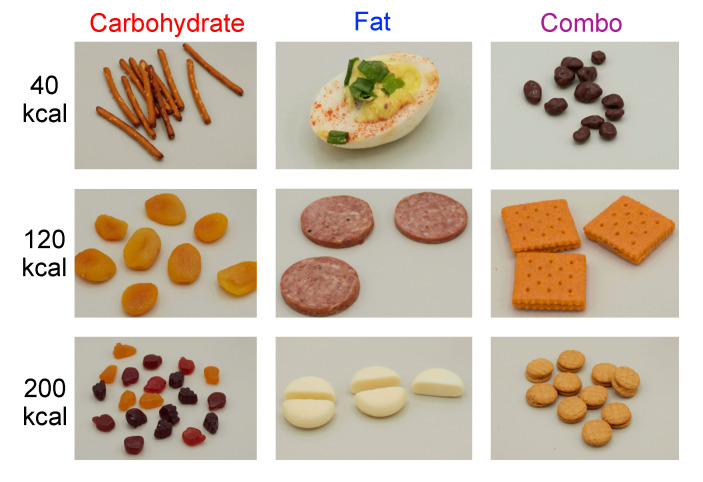
Example food images in the modified MacroPics set [42]. In this set, 4 foods each in the carbohydrate, fat, and combo categories are pictured in the following portions: 40, 120, or 200 kcal.

**Figure 7 nutrients-13-01203-f007:**
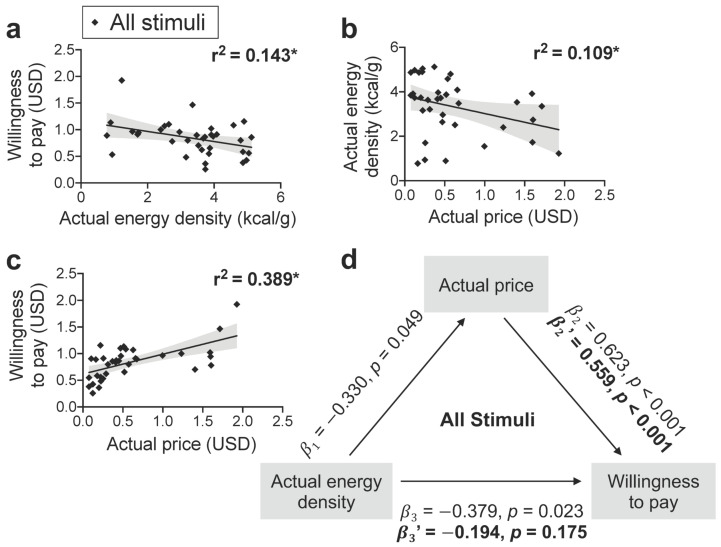
Food price mediates the negative association between WTP and actual energy density when portion size is varied. (**a**,**b**) Actual energy density was negatively related to WTP and actual price for foods pictured in varying portions (40, 120, or 200 kcal). (**c**) Actual price was also positively associated with WTP. (**d**) As the standardized betas (β_1_, β_2_, β_3_) for the separate regressions between energy density, price, and WTP were each significant, we tested whether price mediates the negative association between energy density and WTP. As predicted, the relationship between energy density and WTP *(*β_3′_*)* was no longer significant when the indirect effect of price was accounted for in the regression model, which itself remained significant *(*β_2′_*)*. Resulting r^2^ values are shown for each regression performed on all food items (averaged across all participants). Each data point depicts a single food item and shading indicates 95% CI. * *p* < 0.05.

**Table 1 nutrients-13-01203-t001:** Questions and anchors for the internal state rating scales.

Internal State	Question	Anchors
Hunger	How hungry do you feel?	Not at all hungry, Extremely hungry
Fullness	How full do you feel?	Not at all full, Extremely full
Thirst	How thirsty do you feel?	Not at all thirsty, Extremely thirsty
Potential to eat	How much do you think you could eat right now?	Not another bite, Extremely large amount
Desire to eat	How much do you want to eat right now?	Not at all, Extremely

All scales were continuous horizontal 260-mm visual analog scales. Ratings were scored as the percentage of scale length at which the participant placed their marker.

**Table 2 nutrients-13-01203-t002:** Questions and labels for the food subjective rating scales.

Subjective Variable	Question	Labels
Liking	How much do you like or dislike this food?	See Lim et al., 2009 [48] ^1^
Familiarity	How familiar is this food?	Extremely unfamiliar, Extremely familiar
Frequency of consumption ^2^	How often do you eat this food?	<1× per month, 2–3× per month, 1–2× per month, 3–4× per week, 5+× per week
Healthiness	How healthy is this food?	Extremely unhealthy, Extremely healthy
Expected satiety	How filling do you expect this food portion to be?	Not filling at all, Extremely filling
Estimated energy content ^2^	How many calories are in this portion?	0, 60, 120, 180, 240
Estimated energy density	How energy-dense is this food?	Extremely low, Extremely high
Estimated price ^2^	What is the grocery store price?	0, 2.50, 5 USD

All scales (other than liking) were continuous horizontal 260-mm visual analog scales with labels on each end. ^1^ Liking was assessed on a 150-mm vertical category-ratio Labeled Hedonic Scale [48]. ^2^ The scales for frequency of consumption, estimated energy content, and estimated price contained additional labels evenly spaced between the anchors.

**Table 3 nutrients-13-01203-t003:** Participant characteristics of the full sample (*n* = 60).

Characteristic (Units)	HW (*n* = 30) Mean ± SD, Range	OW/OB (*n* = 30) Mean ± SD, Range
Sex	15 Male, 15 Female	15 Male, 15 Female
Age (yr)	23.6 ± 4.5, 18–34	24.7 ± 4.5, 18–37
Education (yr)	15.3 ± 1.8, 12–20	15.7 ± 2.0, 12–20
Household income ^1^	5.6 ± 1.9, 3–8	4.7 ± 1.8, 1–8
Height (m)	1.70 ± 0.08, 1.59–1.91	1.69 ± 0.08, 1.58–1.90
Weight (kg) *	63.46 ± 8.66, 49.15–83.25	83.95 ± 14.57, 65.50–125.15
Body mass index (kg/m^2^) *	21.92 ± 1.77, 19.13–24.97	29.42 ± 4.44, 25.13–39.96
Waist-hip ratio *	0.81 ± 0.06, 0.68–0.90	0.88 ± 0.06, 0.77–0.99
Body fat (%) *	21.0 ± 8.3, 4.0–34.5	32.6 ± 7.71, 18.8–52.1

^1^ Household income was dummy coded from 1–8 according to 2018 US Census Bureau income percentiles. * *p* < 0.001 (unpaired, two-sample *t*-tests comparing healthy weight (HW) and over-weight/obesity (OW/OB) groups).

## Data Availability

The data presented in this study are available in the Appendix A or upon request from the corresponding author.

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
