# Peer review of "Fat and Carbohydrate Interact to Potentiate Food Reward in Healthy Weight but Not in Overweight or Obesity"

_nutrients, 2021, doi:10.3390/nu13041203_

Round 1
Reviewer 1 Report
This study from the US, aimed to detect if fat and carbohydrate interact to potentiate willingness to pay for snack foods and if estimated, energy density drives this reinforcement signal
Overall, this is a fantastic and well written manuscript, which focus on a relative new area of selecting food, according to body weight status.
Please check again the abstract section and remove all references
The introduction section is well written , and so is the methods section.
Results are well written also and easy to follow.
The Discussion section is very interesting and indicates that fat and carbohydrate intake have a significant effect on normal weight participants.
Congratulations.
Reviewer 2 Report
The aim of the present study was to investigate whether the willingness to pay (WTP) for foods high in fat, high in carbohydrate or combined, differs between individuals with healthy weight (HW) and individuals with overweight/obesity (OW/OB). Furthermore, the authors sought to examine whether, in a US population, they could replicate previous work demonstrating that a combination of fat and carbohydrate potentiated the WTP in individuals with HW and that WTP was positively associated with actual, rather than estimated, energy density of food items. The main findings were that fat and carbohydrate potentiated WTP in individuals with HW but not OW/OB. Additionally, a contrasting (and inverse) relationship was found between WTP and actual energy density which the authors found to be mediated by food price. I would like to commend the authors on a well-designed and well-presented study with a comprehensive analysis of results. Whilst some of the findings were potentially unexpected, the authors provide further exploration of the results and suggest plausible explanations for these findings. Consequently, I recommend that the manuscript is accepted subject to minor revisions. Please see the comments below which would strengthen the manuscript for publication:
- It is currently unclear in the introduction what the authors would expect the differences in WTP responses to be between HW and OW/OB individuals. For example, the research described in paragraph 2 of the introduction may lead readers to expect that the supra-additive effect of fat and carbohydrate would be stronger in individuals with OW/OB as it is suggested that their combination in processed foods may promote their overconsumption (and hence contribute to obesity). However, in the final section of the introduction where the aims are stated, the authors include literature suggesting that a potential insensitivity may be present in obesity leading them to believe that this may influence the impact of the different macronutrients. This is similar in the discussion where rodent literature is discussed in Section 4.1 suggesting the combination of fat and carbohydrate may promote overeating, and then the insensitivity in overweight/obesity is discussed in 4.2. What did the authors hypothesise about the differences in WTP/food reward between individuals with HW and OW/OB before the study was conducted? Stating these hypotheses alongside the aims of the study and addressing the expected direction of the findings in Section 4.2 of the discussion would make this clearer to the reader.
- The aims/purpose of the study in the final paragraph of the introduction and first paragraph of the discussion could be reordered to emphasise the novelties of the study. The title focuses on the key finding of the differences in food reward between HW and OW/OB, however, the two aforementioned sections lead with the replication aspect of the study which slightly undervalues the additional knowledge that the study is adding to the literature. The comparison between HW vs. OW/OB and the application to a US population should be highlighted more as the novelties of the study.
- The power calculation suggested a sample size of 30 participants for each of the two study groups, however, a total of 65 participants were recruited in the study – what was the reasoning for this? Was a target of 65 set prior to the study to account for potential drop out or were an extra 5 participants recruited at the end of the study as the data for 5 existing participants were not sufficient? This could be made clearer in the manuscript.
- The previous study conducted by the authors (DiFeliceantonio et al. 2018, Cell Metabolism) required participants to be fasted and provided participants with a standardised meal prior to the main testing session, however, the present study only required participants to be fasted for 4 hours prior to the main testing session. Why was this chosen rather than a longer fast and standardised meal?
- Table 3 – Given that the results section describes statistical comparisons in participant characteristics between the two groups, symbols denoting significant differences between participant characteristics should be added to Table 3. The table would also look neater/clearer if the breakdown of race and ethnicities were described in the text as opposed to the table.
- Figure 2 – Usually individual data points relate to individual participants rather than individual food items. What is the authors’ rationale for plotting individual food items rather than individual participants?
7. Small typographical error on line 517 in the discussion – the word ‘confound’ should be replaced with ‘confounder’ or ‘confounding variable’.
